

# Tackling the SARS-CoV-2 main protease using hybrid derivatives of 1,5-disubstituted tetrazole-1,2,3-triazoles: an in silico assay

Carlos J. Cortés-García[1,*], Luis Chacón-García[1],
Jorge Emmanuel Mejía-Benavides[2] and Erik Díaz-Cervantes[3,*]

[1] Laboratorio de Diseño Molecular, Instituto de Investigaciones Químico-Biológicas, Universidad Michoacana de San Nicolás de Hidalgo, Morelia, Michoacán, México
[2] Departamento de Enfermería y Obstetricia, Centro Interdisciplinario del Noreste (CINUG), Universidad de Guanajuato, Tierra Blanca, Guanajuato, México
[3] Departamento de Alimentos, Centro Interdisciplinario del Noreste (CINUG), Universidad de Guanajuato, Tierra Blanca, Guanajuato, México
* These authors contributed equally to this work.

## ABSTRACT

In regard to the actual public health global emergency and, based on the state of the art about the ways to inhibit the SARS-CoV-2 treating the COVID19, a family of 1,5-disubstituted tetrazole-1,2,3-triazoles, previously synthesized, have been evaluated through in silico assays against the main protease of the mentioned virus (CoV-2-M$^{Pro}$). The results show that three of these compounds present a more favorable interaction with the selected target than the co-crystallized molecule, which is a peptide-like derivative. It was also found that also hydrophobic interactions play a key role in the ligand-target molecular couplings, due to the higher hydrophobic surfaces into the active site. Finally, a pharmacophore model has been proposed based on the results below, and a family of 1,5-DT derivatives has been designed and tested with the same methods employed in this work. It was concluded that the compound with the isatin as a substituent (P8) present the higher ligand-target interaction, which makes this a strong drug candidate against COVID19, due can inhibit the CoV-2-M$^{Pro}$ protein.

Corresponding author
Erik Díaz-Cervantes, e.diaz@ugto.mx

## INTRODUCTION

Recently, a new kind of coronavirus strain was discovered in Wuhan city in Hubei province, central China. This virus is known as severe acute respiratory syndrome coronavirus 2 (SARS-CoV-2) and has caused the coronavirus disease 2019 (COVID-19), which has now become a pandemic threat (*Gabutti et al., 2020*; *Gralinski & Menachery, 2020*; *Jin et al., 2020*). At present, SARS-CoV-2 has caused thousands of deaths and more than 5 million people have been infected worldwide, becoming a global public health emergency (*Sohrabi et al., 2020*). Despite the fact that there are no specific antivirals to

treat the COVID-19, the scientific community is using the drug repurposing of some FDA approved drugs such as lopinavir, remdesivir and chloroquine as a rapid strategy to find a cure (*Kandeel & Al-Nazawi, 2020*; *Li & De Clercq, 2020*; *Li et al., 2020*; *Shah, Modi & Sagar, 2020*). Nonetheless, the need to develop a new specific antiviral drug is still urgent.

A quick and efficient way to find a new drug candidate is through computer-aided drug design (CADD) which is a powerful tool used to find new compound by reducing risk, time and cost of research in the drug discovery process (*Baig et al., 2016*; *Bisht & Singh, 2018*; *Ferreira et al., 2015*; *Hoque et al., 2017*). Moreover, in February 2020, the first high-resolution crystal structure of the main protease of SARS-CoV-2 was published (PDB code: 6lu7) (*Liu et al., 2020b*). Such protein is essential in the virus life cycle, thus making it a key target in the quest of developing novel antiviral agents.

Moreover, regards other studies of the main protease of SARS-CoV-2 (CoV-2-M$^{Pro}$), which presents a similar structure to the M$^{Pro}$ of the SARS-CoV (*He et al., 2020*; *Liu et al., 2020a*; *Vellingiri et al., 2020*), has been reported that the hydrogen bonds play a key role in the ligand-target interactions (*Chou, Wei & Zhong, 2003*), as well as in the present year have been perform several studies about the interactions of organic molecules with the CoV-2-M$^{Pro}$ (*Peele et al., 2020a*; *Abraham Peele et al., 2020b*). Also, according to the state of the art concerning potential candidates that target this specific enzyme, highlights 1,2,3-triazoles, synthesized by the Dehaen group, were proposed as potential anti-coronavirus agents (*Karypidou et al., 2018*). In this context, the present work aimed to evaluate in silico a series of 1,5-disubstituted-tetrazole derivatives which were previously synthesized in our research group (*Aguilar-Morales et al., 2019*) and whose biological and theoretical essays have not been reported.

Recent reports have shown the plethora of molecules that can interact with the selected target (CoV-2-M$^{Pro}$); ranging from alkaloids found in medicinal plants (*Qamar et al., 2020*), as well as in other plants with known health properties, as garlic (*Phuong-Thuy et al., 2020*), to the use of alpha ketoamide (*Zhang et al., 2020*). Based on this context, the present work proposes several 1,5-disubstituted tetrazole-1,2,3-triazoles as some novel plausible inhibitors of the CoV-2-M$^{Pro}$. The above-mentioned compounds were evaluated through *docking* assays to obtain the target-ligand interactions that take place. Additionally, a pharmacophore model was performed based on the target (considering the electrostatic, hydrophobic and hydrogen bond interactions) leading to the to design of a novel family of molecules that can potentially inhibit the CoV-2-M$^{Pro}$.

## COMPUTATIONAL METHODS

The Cartesian coordinates from the selected target, CoV-2-M$^{Pro}$, were obtained from the protein data bank (PDB code: 6lu7), which was one of the first crystallized structures of the main protease of the SARS-CoV-2 virus. Furthermore, the Chimera package was used to add charges, remove solvents and correct residues of the target structure (*Pettersen et al., 2004*).

Moreover, the 1,5-disubstituted tetrazole-1,2,3-triazoles and the co-crystallized compound (into the CoV-2-M$^{Pro}$ structure), see Fig. 1, which are considered as ligands,

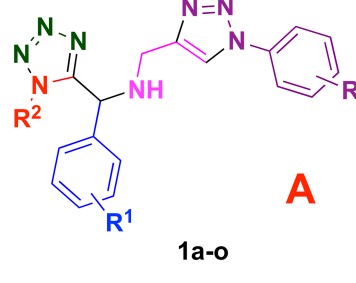

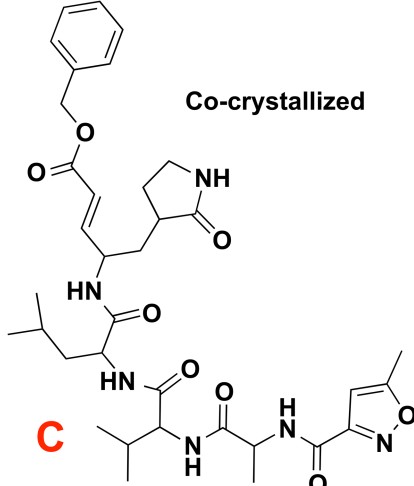

| Compound | R¹ | R² | R³ |
|---|---|---|---|
| 1a | H | t-Bu | 4-CN |
| 1b | 4-F | t-Bu | 4-CN |
| 1c | 4-Br | Cy | 4-CN |
| 1d | 4-OMe | t-Bu | 4-CN |
| 1e | 4-OMe | Cy | 4-CN |
| 1f | 2-Cl | t-Bu | 4-CN |
| 1g | 2-Br | Cy | 4-CN |
| 1h | 2,4,5-triMe | t-Bu | 4-CN |
| 1i | H | Cy | 4-Cl |
| 1j | 4-F | t-Bu | 4-Cl |
| 1k | 4-Br | t-Bu | 4-Cl |
| 1l | 4-OMe | t-Bu | 4-NHCOMe |
| 1m | 4-OMe | Cy | 4-NHCOMe |
| 1n | 4-Br | t-Bu | 2-COPh-4Cl |
| 1o | 2-F | Cy | 2-COPh-4Cl |

**Figure 1 Modeled co-crystallized and 1,5 Disubstituted tetrazole 1,2,3-triazoles.** (A) 1,5 Disubstituted tetrazole 1,2,3-triazoles, (B) their R's substituents, and (C) the molecule co-crystallized in the selected target.

were modeled using the Avogadro software (*Jin et al., 2020*) and charged with the Chimera package (*Pettersen et al., 2004*). However, to obtain a better approach, the ligands were optimized at the UFF level (*Rappe et al., 1992*) using the Gaussian 09 (G09) package (*Frisch et al., 2009*). Note that, the UFF method was employed due to the good distance and angle bonds it provides to organic molecules, as well as the low computational cost.

Ligand-target interactions were obtained using the Molegro MVD package and the in silico molecular couplings, so-called molecular *docking*, were performed through the MolDock score function (*Thomsen & Christensen, 2006*). Also, the electrostatic, hydrogen bond and hydrophilic surfaces interactions were obtained with the same MVD software. Although several methods are frequently reported when studying ligand-target interactions such as the DFTB (*Allec et al., 2019*; *Morao et al., 2017*), the selected method was used as a preliminary approach to evaluate these interactions.

Finally, the pharmacophore model was developed using the ZINCPharmer server (*Koes & Camacho, 2012*), considering the obtained properties obtained throughout the whole study. It is noteworthy to mention that the proposed novel inhibitors underwent the same process as the 1,5-disubstituted tetrazole-1,2,3-triazoles and were evaluated using the selected target with the above method.

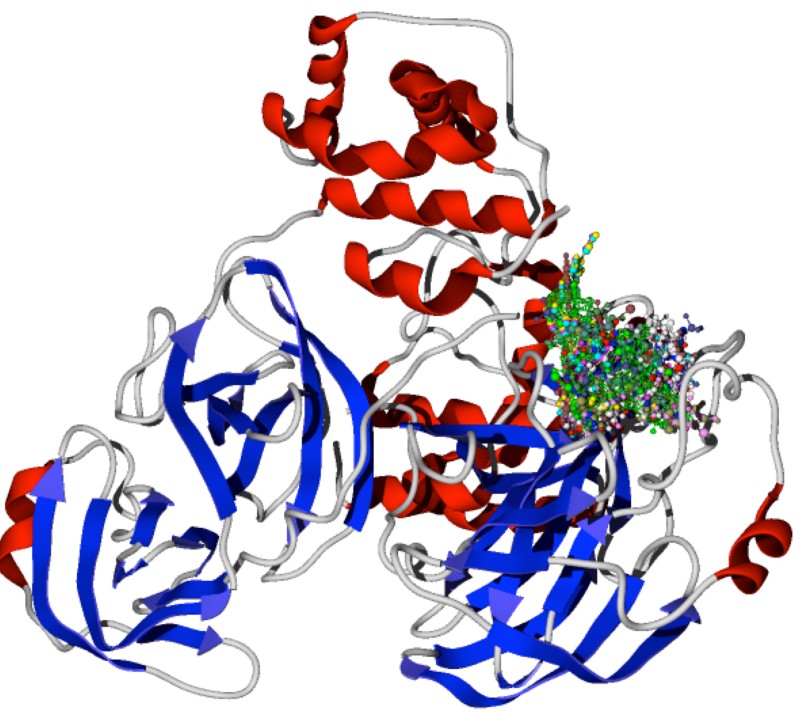

**Figure 2** *Docking* **of the 1,5-disubstituted tetrazole-1,2,3-triazoles with the protein CoV-2-M^Pro.**

## RESULTS

### Molecular docking

The specific docking of all the modeled ligands, see Fig. 1, into the active site of the protein CoV-2-M^Pro is shown in Fig. 2. It is evident that all the ligands docked similarly to the selected target and interacted with the catalytic triad residues, such interaction will be explained in the discussion section.

Table 1 shows the hydrogen bond energies, the electrostatic interactions and the LE values obtained for the selected ligands. Additionally, Fig. 3 shows the active site of the target, with the co-crystalized ligand and two of the best ligands interacting.

### Hydrogen bonds, electrostatic interactions and hydrophobic interactions

The hydrogen bond and electrostatic interactions between the selected target and the ligand 1e, which is the compound with the most favorable ligand-target interactions, are depicted in Fig. 4. The principal interactions are those with histidine residues, as well as one with serine and glutamine.

The hydrophobicity surfaces of the active site show that in the deep zone of the cavity a hydrophobic zone can be found, as shown in Fig. 5 in blue colored surfaces. However, the front of the cavity and one site in the upper-left zone shows a higher hydrophilic site, depicted in red color in Fig. 5.

**Table 1 Interaction energies between the modeled ligands and the protein CoV-2-M^Pro.** All the units are represented in kcal/mol. $H_{Bond}$ means the hydrogen bond interactions, Elstat the electrostatic energies, VdW indicates the Van der Waals energy and, LE is the ligand efficiency (LE = $E$/No heavy atoms).

| Molecule | $E$ | $H_{Bond}$ | Electro | VdW | LE |
|---|---|---|---|---|---|
| P8 | −255.79 | −7.99 | −0.11 | −60.73 | −5.44 |
| P10 | −210.29 | −8.31 | 0.74 | −56.38 | −5.26 |
| 1e | −181.75 | −2.52 | −0.65 | −52.43 | −5.19 |
| P7 | −212.54 | −6.86 | 0.05 | −51.71 | −5.18 |
| 1h | −174.15 | −5.64 | −0.45 | −49.54 | −5.12 |
| P9 | −213.61 | −11.93 | −0.38 | 22.42 | −5.09 |
| P6 | −198.34 | −8.17 | 0.03 | −54.14 | −5.09 |
| P5 | −198.33 | −5.24 | −0.72 | −49.06 | −5.09 |
| 1k | −157.03 | −8.32 | −0.54 | −22.57 | −5.07 |
| Co-crystal | −211.79 | −8.38 | −0.61 | −54.52 | −5.04 |
| 1a | −155.38 | −6.86 | −0.10 | −42.84 | −5.01 |
| P1 | −215.37 | −11.53 | 1.72 | −66.12 | −5.01 |
| 1j | −155.23 | −5.05 | −0.42 | −44.01 | −5.01 |
| 1g | −169.91 | −6.86 | 0.58 | 53.31 | −5.00 |
| 1i | −159.14 | −10.08 | 0.09 | −26.84 | −4.97 |
| P2 | −213.91 | −7.78 | 0.82 | −34.42 | −4.86 |
| 1m | −178.32 | −0.82 | −0.78 | −38.11 | −4.82 |
| P3 | −201.55 | −13.36 | −0.37 | 62.82 | −4.80 |
| 1o | −193.90 | −9.15 | −0.09 | −51.42 | −4.73 |
| 1f | −150.08 | −6.83 | 0.57 | −28.37 | −4.69 |
| P4 | −195.21 | −6.29 | −1.00 | −59.43 | −4.65 |
| 1b | −147.94 | −10.63 | −0.13 | −38.52 | −4.62 |
| 1c | −156.95 | −12.12 | −0.05 | −39.49 | −4.62 |
| 1d | −151.19 | −11.98 | −0.13 | −40.69 | −4.58 |
| 1n | −177.09 | −6.68 | −0.57 | −26.16 | −4.54 |
| 1l | −151.39 | −9.55 | −0.03 | −28.60 | −4.45 |

## Pharmacophore model

Taken into account the results gathered, especially the docking results obtained by both, the 1e ligand and the co-crystallized ligand, and taken the hydrophobic surfaces as one of the main interactions, a pharmacophore model was developed. The principal aim in developing this model was to predict and propose the molecular fragments that are necessary to interact with CoV-2-M^Pro, which would result in virus inhibition.

The proposed pharmacophore model is shown in Fig. 6, and consists of 10 principal components: two hydrophobic fragments (Hy, depicted in green color), one aromatic fragment (Ar, colored in blue color), three hydrophobic–aromatic fragments (Hy–Ar, represented in purple color), two hydrogen donor fragments (HD, depicted in gray color) and two hydrogen acceptor fragments (colored in orange color).
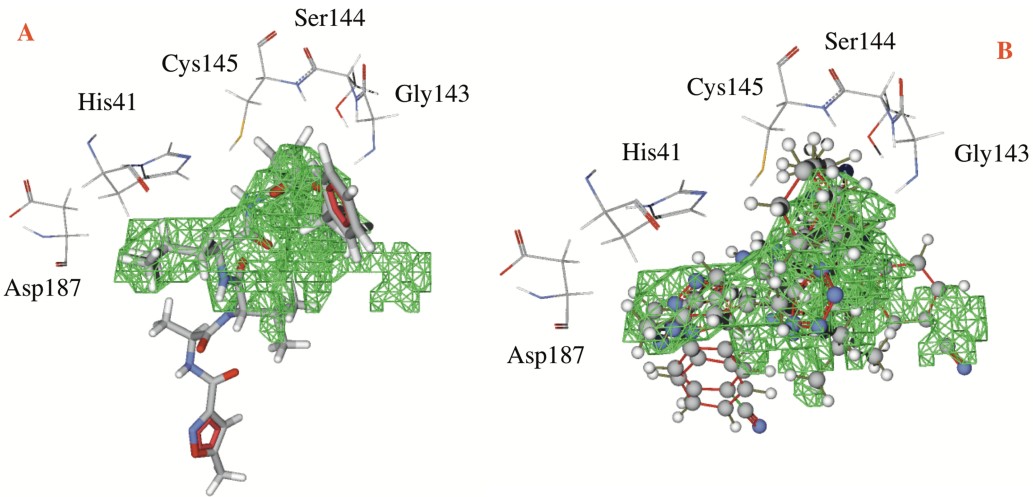

**Figure 3 Catalytic site of the CoV-2-M<sup>Pro</sup> protein.** (A) The co-crystalized ligand, and (B) the two best ligands interacting into the catalytic site of the Cov-2-M<sup>Pro</sup> protein.

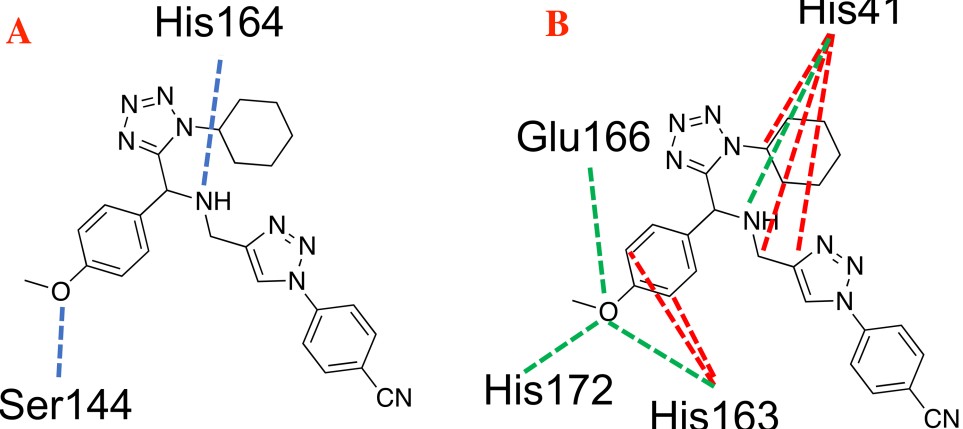

**Figure 4 Main non-covalent interactions between the CoV-2-M<sup>Pro</sup> protein and the best ligand 1e.** (A) Hydrogen bond, and (B) electrostatic interactions; Dotted blue lines represent the hydrogen bond interaction, as well as dotted green lines, represent the attractive electrostatic interactions and dotted red lines the repulsive electrostatic interactions.

## Proposing molecules

Based on the obtained results and considering the pharmacophore model, a series of ten 1,5-disubstituted tetrazole-1,2,3-triazoles have been proposed as inhibitors of the CoV-2-M<sup>Pro</sup>, which are depicted in Fig. 7.

Furthermore, Fig. 8A shows that that molecule P8 prefers to interact in the right side of the molecule in a similar manner as the co-crystallized ligand and 1e compounds. The structure of compound P8 is located in the deep of the active site.

Finally, Fig. 9 shows the molecules 1e and P8 in the pharmacophore model, and reveals the occupied space by these molecules into the pharmacophore model.
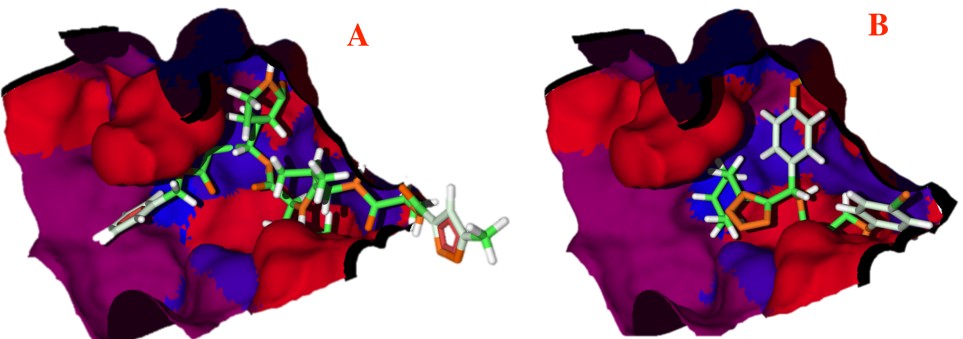

**Figure 5 Hydrophobic interactions between the CoV-2-M^Pro protein and the main ligands.** Hydrophobic interactions between the Cov-2-M^Pro protein and (A) the co-crystallized ligand, and (B) 1e. Blue surfaces represent hydrophobic sites, red surfaces are hydrophilic zones.

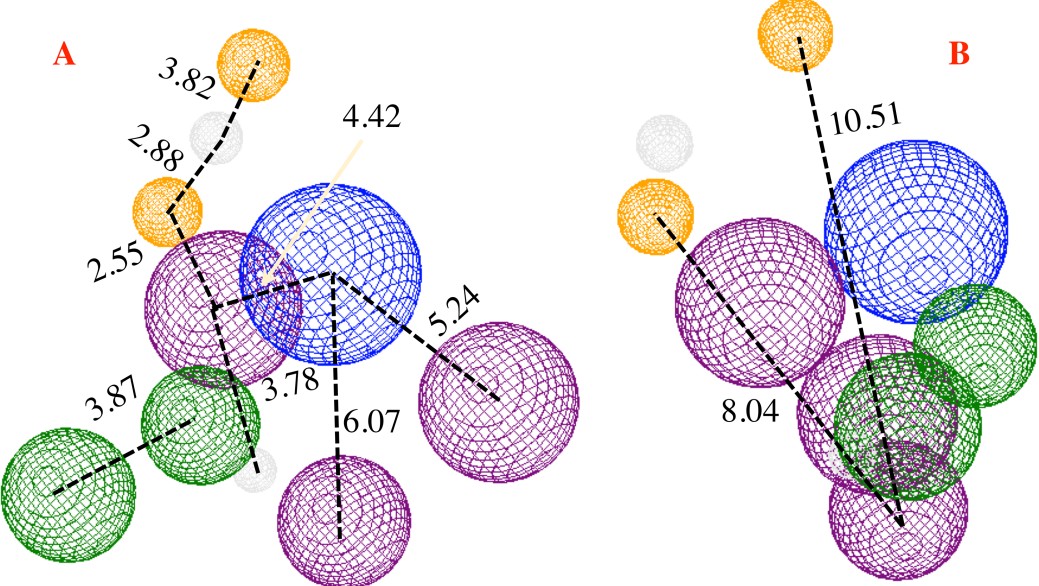

**Figure 6 Pharmacophore model for the CoV-2-M^Pro protein.** (A) Front and (B) lateral view for the pharmacophore model for the Cov-2-M^Pro protein. Green spheres depict the hydrophobic segments (Hy), blue spheres are the aromatic segments (Ar), purple spheres represent the hydrophobic and aromatic segments (Hy–Ar), gray spheres depict the hydrogen donor segments (HD) and the orange sphere represents the hydrogen acceptor segments (HA).

## DISCUSSION

### Molecular docking

To evaluate the best ligand docked in the selected target, CoV-2-M^Pro, the MolDock score energy was considered as a parameter of measurement. Furthermore, the ligand efficiency (LE = Energy/No. of heavy atoms) was used to determine with better precision the ligand-target binding strength. This parameter gives the energy provided per atom in the ligand-target interaction, making it a better way of comparison between ligands with different number of atoms, see Table 1.

**Figure 7 Designed compounds that present favorable interactions with the CoV-2-M^Pro protein.**
(A–J) Compounds presenting structural variation in their R's substituents.

In comparison with the co-crystallized ligand reported in the PDB file, 1e is 0.04 kcal/mol more stable (*Liu et al., 2020b*). Also, P8 is the molecule with highest ligand-target interaction energy.

Moreover, the hydrogen bond ($H_{bond}$), electrostatic (Elstat) interactions and the Van der Waals energies (VdW) are shown in Table 1, demonstrating that the VdW the limiting energy in obtaining a better ligand-target interaction for this kind of systems.

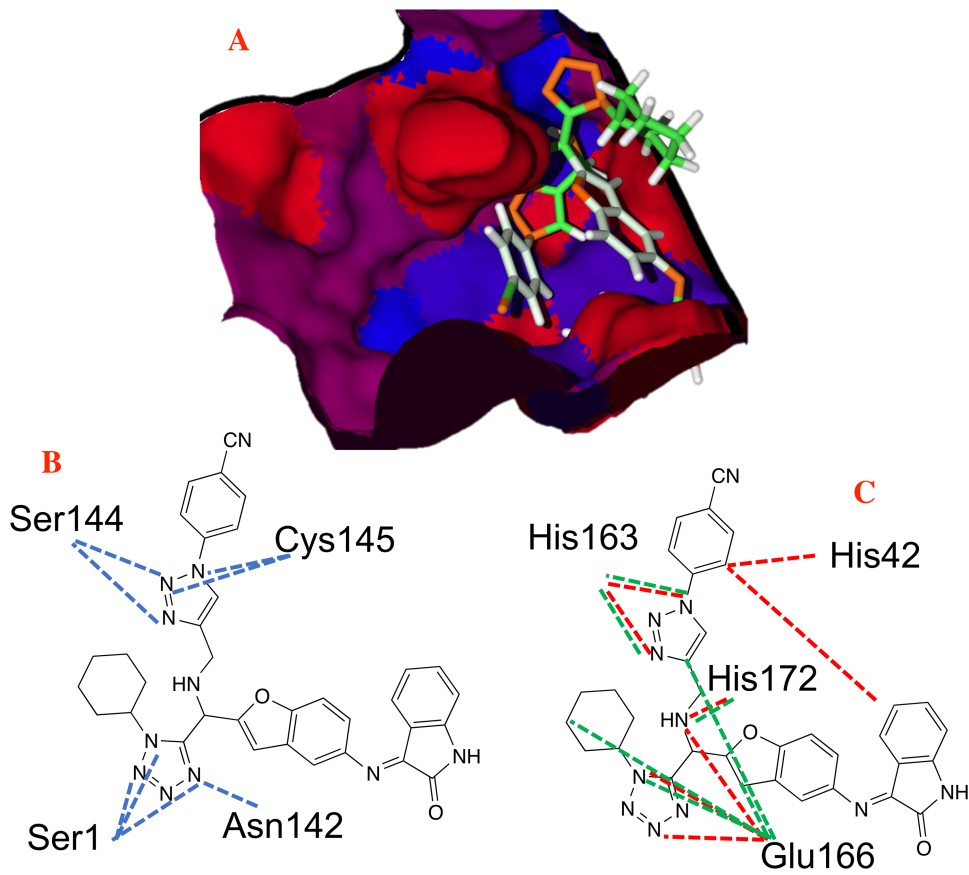

**Figure 8 Non-covalent interactions between p8 and the CoV-2-M^Pro protein.** (A) Hydrophobic, (B) Hydrogen bond and (C) electrostatic interactions between p8 and the Cov-2-M^Pro protein. Blue surfaces represent hydrophobic sites, red surfaces are hydrophilic zones, dotted blue lines represent the hydrogen bond interactions, as well as the dotted green and red lines are the attractive and repulsive interactions, respectively.           

In other hand, best ligands show the most favorable VdW energies. However, the state of the art regarding this protease indicates that the hydrogen bond interactions are one of the most important energies, especially with the amides of the catalytic triad residues (Gly143, Cys145 and Ser144) (*Zhang et al., 2020*). Note that Table 1 includes compounds P1-10, which are the designed potential inhibitors presented in this work and will be boarded in "Proposing Molecules".

In respect to the docked cavity, our results are in line with the results obtained by other authors, and show the interactions of the ligands into the active site of the protein, in the so-called catalytic triad (Gly143, Ser144 and Cys145), see Fig. 3. At the same time, the His41 and the Asp187 are important residues in the onset of the electronic transfer, which is the key mechanism in peptide bond rupture for this kind of protease. Other key fragments in the oxyanion hole includes the Gly143 and Ser144 residues, which according to Warshel and coworkers (*Kamerlin, Chu & Warshel, 2010*; *Mukherjee & Warshel, 2012*), stabilize the anionic intermediary. This helps identify the active site and is a way to corroborate that this cavity is the target site of the protein.

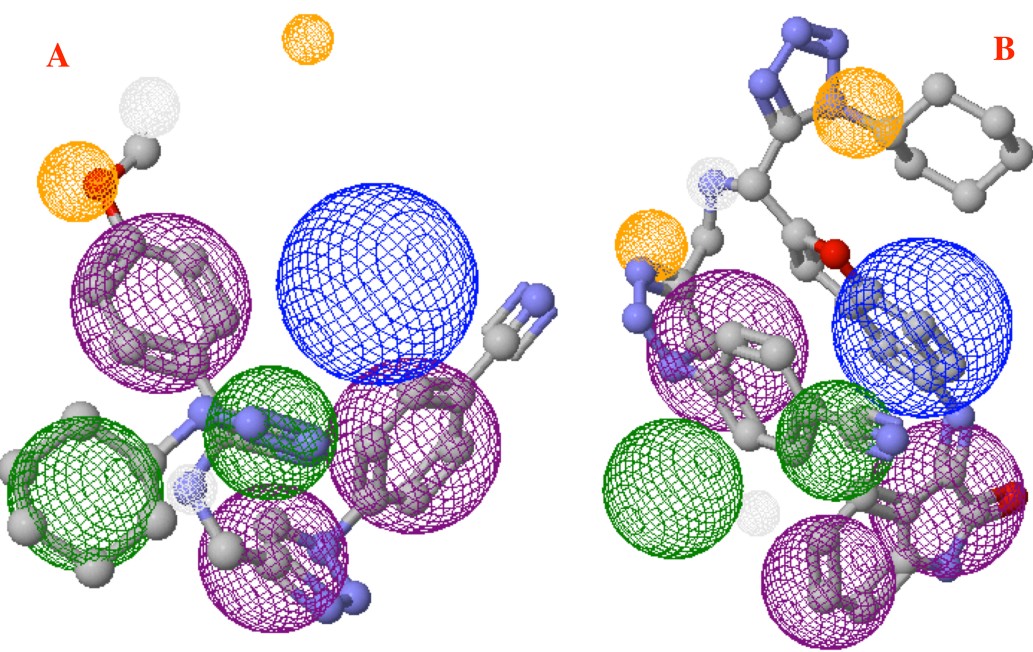

**Figure 9 Main molecules overlap into the pharmacophore model for the CoV-2-M^Pro protein.** (A) 1e and (B) p8 molecules overlap into the pharmacophore model for the CoV-2-M^Pro protein. Green spheres depict the hydrophobic segments (Hy), blue spheres are the aromatic segments (Ar), purple spheres represent the hydrophobic and aromatic segments (Hy–Ar), gray spheres depict the hydrogen donor segments (HD) and the orange sphere represents the hydrogen acceptor segments (HA).

Figure 3 shows first the co-crystallized molecule (a peptide like derivative) which directly interacts with the catalytic triad and part of the molecule dock perfectly in almost the whole cavity. Furthermore, Fig. 3B depicts the two best ligands interacting in a similar manner to the co-crystallized molecule, but filling the right site of the computed cavity. This behavior might explain the most favorable interactions seen when comparing them with the co-crystallized molecule, see Table 1.

## Hydrogen bonds, electrostatic interactions and hydrophobic interactions

The 1e ligand presents hydrogen bond interactions with Ser144 and His164, see Fig. 4A. Note that these are residues of the catalytic site. Furthermore, Fig. 4B shows the same ligand interacting with His41, His163, Glu166 and His172 via electrostatic forces, in which the repulsive electrostatic interactions are more prominent than the attractive ones. This, as a result of the similar partial charge (positive–positive or negative–negative) between 1e and histidine residues. This kind of interaction plays a key role in the final conformation of the bioactive posse. Generally, the electrostatic interactions are one of the limiting energies in the ligand-target coupling and ligand 1e shows a value of −0.65, being this one of the highest ones of the table.

To better evaluate the ligand-target interactions, it is necessary to carry out an analysis of the hydrophobic surfaces of the selected cavity. Figure 5A shows the interactions

between the co-crystallized ligand and the selected target, from a hydrophobic behavior. The ligand takes a conformation into the cavity occupying only the deep zone of the active site (the more hydrophobic site) and avoids the interactions with the front site of the cavity. Note that ligand 1e shows a similar interaction with the active site, see Fig. 5B.

Contrary to the co-crystallized ligand, compound 1e does not cross the cavity space in the right site. However, in interacts in the deep site of the cavity, docking less in the left site of the surface. Note that ligand 1e presents higher L.E., than the co-crystallized ligand, see Table 1. Therefore, it is clear that the presence of aromatic and hydrophobic rings in both molecules is essential and key for better interactions. The hydrophilic interactions are practically negligible.

## Pharmacophore model

Analyzing Fig. 6, it is clear that the low side of the model needs mostly hydrophobic fragments, and in the top, the fragments are HD and HA. The volume of the pharmacophore model makes us think that to tackle the $M^{Pro}$ of the SARS-Cov-2 virus, it is necessary a molecule with rings along their whole structure.

Highlights that with the proposed pharmacophore model can be design a family of compounds which can inhibit the $M^{Pro}$, avoiding the virus replication and promoting the cure for the COVID19.

## Proposing molecules

Based on the obtained results and considering the pharmacophore model, a series of ten 1,5-disubstituted tetrazole-1,2,3-triazoles have been proposed as inhibitors of the CoV-2-$M^{Pro}$, which are depicted in Fig. 7. As shown in Table 1, the P8 and P10 designed molecules present the more favorable interactions with the selected target as they show a more negative LE value than the other evaluated compounds. Note that, six of the ten designed molecules exhibit better interaction with the CoV-2-$M^{Pro}$ than the co-crystallized molecules, which is the reference molecule.

On the other hand, the compound P8 has an isatin scaffold (1H-indole-2,3-dione) as part of its structure, which is considered a privileged structure given its broad biological and pharmacological activity. Some of which include antibacterial, anticancer, antitubercular, antimalarial, antifungal, anticancer, anti-HIV and in general antiviral (*Varun, Sonam & Kakkar, 2019*). Analyzing the bio-active possess of compound P8, it is seen that it promotes an intramolecular stabilization due to two stacking interactions: one with the triazole ring and the other with the benzene ring, face-to-edge and face-to-face, respectively.

Figure 8 shows the principal hydrophobic interactions between P8 and the Cov-2-$M^{Pro}$, depicted in blue surfaces. These results can be explained by the higher quantity of rings in the P8 structures, which could promote hydrophobic interactions.

In the case of the hydrogen bond interactions, molecule P8 interacts not only with the catalytic triad, specifically with the Ser144 and Cys145, and presents a higher number of interactions with other residues that include Ser1 and Asn142. The last one promotes

a higher $H_{bond}$ energy than compound 1e, see Table 1. In fact, (*Liu et al., 2020b*) mention that hydrogen bond interactions play a key role in the ligand-target interaction as highlighted in the state of the art. Moreover, the electrostatic interactions between P8 and the selected target take place at the residues His41, His163, Glu166 and His172 residues, which in terms of Elstat energy, do not promote a favorable interaction (To understand better the behavior of the electrostatic interactions as a function of the different moieties in the studied molecules, the molecular electrostatic potential surfaces are depicted in Supplemental Files).

On the other hand, molecule 1e, which was previously synthesized by some of us, and the designed compound P8 were evaluated into the pharmacophore model and analyzed through the segments docked with the proposed structure. Figure 9A shows the molecule 1e in the pharmacophore model, which reveals that this molecule needs some components to complete all the pharmacophore fragments. Specifically, it needs an aromatic moiety, as well as an HD and HA fragments in the top of the molecule.

Finally, Fig. 9B depicts the P8 structure into the pharmacophore model and shows that this molecule only an Hy and one HD fragments in order to complete all the requirements. Analyzing the results, it is clear that to obtain some better molecules that could inhibit the Cov-2-M$^{Pro}$ it is necessary to have a system that includes some rings in their structure. Also, the right side is the more important site of the cavity, as long as the size of the molecule does not overpass the size of the cavity.

## CONCLUSIONS

A family of compounds previously synthesized by some of us was tested to inhibit the protein Cov-2-M$^{Pro}$, the results show that three of these compounds present a more favorable interaction with the selected target than the co-crystallized molecule, which is a peptide-like derivative. Moreover, although the fact that hydrogen bond interactions are mentioned in the state of the art about the selected protease, it can also be found that the electrostatic interactions and main the hydrophobic interactions play a key role in the ligand-target molecular couplings.

At the same time, the results reveal that a molecule can couple into the active site, which presents higher hydrophobic surfaces. Thus, in the quest to develop potential candidates it is essential to synthesize some molecules with a higher number of aromatic rings in their structures. Note that the residues of the active site interact in a stronger way with the best coupled ligand.

Finally, a pharmacophore model has been designed and used to propose a new family of 1,5-disubstituted tetrazole-1,2,3-triazoles derivatives. These compounds are potential candidates to be synthesized as a perspective of this work. Based on the obtained results, the best ligands were coupled with the pharmacophore model, highlights the importance of the isatin moiety. Also, the pharmacophore model revealed that derivatives bearing the isatin substituent have a higher potential in the design of new drugs against the SARS-Cov-2. Hydrophobic and stacking interactions also play a key role in the design of new drug candidates to treat the COVID19.

## ACKNOWLEDGEMENTS

We are grateful to the Laboratorio Nacional de Caracterización de Propiedades Fisicoquímicas y Estructura Molecular (UG-UAA-CONACYT, Project: 123732) for the computing time provided. Erik Díaz-Cervantes acknowledges Citlalli, Emily, and Naomi for being part of the writing process.

### Funding

Erik Díaz-Cervantes received support from the SICES in the project: IJ-19-77 (Programa de empuje científico y tecnológico modalidad "Apoyo a Investigadores Jóvenes").
The funders had no role in study design, data collection and analysis, decision to publish, or preparation of the manuscript.

### Grant Disclosures

The following grant information was disclosed by the authors:
SICES in the project: IJ-19-77 (Programa de empuje científico y tecnológico modalidad "Apoyo a Investigadores Jóvenes").

### Competing Interests

The authors declare that they have no competing interests.

### Author Contributions

- Carlos J. Cortés-García conceived and designed the experiments, performed the experiments, analyzed the data, authored or reviewed drafts of the paper, and approved the final draft.
- Luis Chacón-García performed the experiments, performed the computation work, authored or reviewed drafts of the paper, and approved the final draft.
- Jorge Emmanuel Mejía-Benavides conceived and designed the experiments, performed the experiments, prepared figures and/or tables, and approved the final draft.
- Erik Díaz-Cervantes conceived and designed the experiments, performed the experiments, analyzed the data, performed the computation work, prepared figures and/or tables, authored or reviewed drafts of the paper, and approved the final draft.

### Data Availability

The raw data is available in the Supplemental File.

### Supplemental Information

Supplemental information for this article can be found online at http://dx.doi.org/10.7717/peerj-pchem.10#supplemental-information.

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
