# Peer review of "Tackling the SARS-CoV-2 main protease using hybrid derivatives of 1,5-disubstituted tetrazole-1,2,3-triazoles: an in silico assay"

_PeerJ Physical Chemistry, doi:10.7717/peerj-pchem.10_

## Round 0.1 · original submission · Minor Revisions

Please address the questions raised by the reviewers. You can ignore their citation requests.

Reviewer 1 ·

Basic reporting

The article meets the standards of this journal.

Experimental design

The experimental design meets the standards of the journal.

Validity of the findings

Suggested improvements are given in the "general comments" section.

Additional comments

In this submission to this PeerJ journal, the authors present a series of calculations on ligand-target interactions using the Molegro MVD package. The authors also investigate in silico molecular couplings, so-called molecular docking using the MolDock score function from Thomsen and co-workers.

I consider this manuscript to be of interest to readers of this PeerJ journal, and I am supportive of publication with a minor note. There has actually be prior work using advanced methods (such as DFTB) for understanding ligand-protein interactions, which should also be mentioned:

J. Comput. Chem. 38, 1987–1990 (2017)
J. Chem. Theory Comput. 15, 2807-2815 (2019)

In particular, these prior works have shown that DFTB is more accurate than MD calculations for ligand-protein interactions, and is faster than DFT for large systems, which should be mentioned as previous studies relevant to this field. With this minor revision, I would be receptive towards publication.

·

Basic reporting

This is highly requested that authors should increase the understandability of the sentences all over the manuscript. In the introduction section please mention the specific benefit for the substitution conducted in this work. In 64 – 67 please mention the abbreviation and please provide information about what kind of biological theoretical understanding is necessary and why it is important and improve the understanding from the previously published works. In the last paragraph of the introduction please arrange relevant information about your study and mention the methodology used in this work. Some of the general issues have to be corrected
Line 44- please update the number according to the present situation.
Line 114 – insert (-) sign
Line 139 – why SER144 is capital?
In table 1 – Please insert the unit accordingly in the main text.
This type of irrelevancy should be carefully removed.

Experimental design

I hope the authors don’t mind including more detailed information about the simulation techniques and the initial setup of the simulation. Further, in line 88 – 89, optimization was conducted at the UFF level, what is the reason for choosing UFF, authors should define the reason and explain the validity of the choosing such methods.

Validity of the findings

Electrostatic charge distribution on the surface of the 1,5 – disubstituted tetrazole – 1,2,3-triazoles is important for this study and read will like to see the distribution of the change which could provide easy understandings, authors can provide such information by making a surface grid figure at least for the co-crystallized, 1e, and P8 compounds.

A clear representation of co-crystalized ligand and the 1e ligand is necessary for easy understandings.

In figure 4 the reason for the repulsive interactions has to be explained, further, it is worthy to provide an explanation about the difference of the interaction energies for different target sites.

·

Basic reporting

This article was well written in english and was clear with out ambiguity . Technicalli it is looking sound and professional.

Experimental design

Methods were described sufficiently and followed simple straight proved and widely accepted protocols which makes consisitent

Validity of the findings

Results section authors have given the name recombinant strain
COV-2MA

Additional comments

Authors required to include the following references at appropriate places in the manuscript

https://www.sciencedirect.com/science/article/pii/S2352914820301805

https://www.tandfonline.com/doi/full/10.1080/07391102.2020.1770127

---

## Round 0.2 · accepted · Accept

We appreciate your submission to PeerJ Phys Chem. Thank you for your patience as it has been unusually difficult to find reviewers during this difficult time.